# Oligomeric state of the ZIKV E protein defines protective immune responses

Stefan W. Metz [1]*, Ashlie Thomas[1], Alex Brackbill[2], John Forsberg[2], Michael J. Miley[2], Cesar A. Lopez[1], Helen M. Lazear [1], Shaomin Tian[1] & Aravinda M. de Silva[1]*

The current leading Zika vaccine candidates in clinical testing are based on live or killed virus platforms, which have safety issues, especially in pregnant women. Zika subunit vaccines, however, have shown poor performance in preclinical studies, most likely because the antigens tested do not display critical quaternary structure epitopes present on Zika E protein homodimers that cover the surface of the virus. Here, we produce stable recombinant E protein homodimers that are recognized by strongly neutralizing Zika specific monoclonal antibodies. In mice, the dimeric antigen stimulate strongly neutralizing antibodies that target epitopes that are similar to epitopes recognized by human antibodies following natural Zika virus infection. The monomer antigen stimulates low levels of E-domain III targeting neutralizing antibodies. In a Zika challenge model, only E dimer antigen stimulates protective antibodies, not the monomer. These results highlight the importance of mimicking the highly structured flavivirus surface when designing subunit vaccines.

[1] Department of Microbiology and Immunology, University of North Carolina at Chapel Hill, Chapel Hill, NC, USA. [2] Department of Pharmacology, University of North Carolina at Chapel Hill, Chapel Hill, NC, USA. *email: swmetz@med.unc.edu; aravinda_desilva@med.unc.edu

Zika virus (ZIKV) is an enveloped single stranded RNA virus that is transmitted by mosquito vectors. As a member of the flavivirus genus, ZIKV shares many structural features with other medically important flaviviruses such as Wes Nile virus, yellow fever virus, and dengue virus (DENV). The flavivirus envelope (E) protein is a major target of neutralizing and protective human antibodies. E oligomers form complex quaternary structure epitopes on the viral surface which are of particular importance as protective human antibodies bind to these higher order protein structures[1].

ZIKV infection during pregnancy can result in neurodevelopmental defects (congenital Zika syndrome). This has stimulated work on subunit vaccines, as live attenuated and other replicating virus vaccines are contraindicated during pregnancy. However, for flaviviruses subunit vaccines based on recombinant E proteins (rE) have performed poorly in preclinical studies compared with virus or virus-like particle (VLP) vaccines[2–6]. In aqueous solution, flavivirus rE proteins are in a dynamic equilibrium that favors the monomer over the dimer at physiological temperature, which likely explains poor binding by potently neutralizing human antibodies targeting quaternary epitopes and the overall poor immunogenicity of these antigens in preclinical studies[2,7,8].

Here, we investigate if ZIKV rE dimers are better subunit vaccine antigens than monomers. Unlike the E monomer, the dimer is recognized by strongly neutralizing mAbs isolated from ZIKV patients and the dimer, not the monomer, stimulates strongly neutralizing and protective antibodies that targeted epitopes that were similar to epitopes recognized by human antibodies following natural Zika virus infection. These results highlight the importance of mimicking the highly structured flavivirus surface when designing subunit vaccines. The flavivirus field has a long history of using E monomers as vaccine antigens with limited success. These results highlight the importance of mimicking the highly structured flavivirus surface when designing subunit vaccines, and are applicable to developing second generation subunit vaccines against Zika as well as other medically important flaviviruses like dengue and yellow fever.

## Results

**Expression of ZIKV rE monomers and homodimers**. Here, to investigate if ZIKV rE dimers are better subunit vaccine antigens than monomers, we expressed ZIKV rE variants (aa1–402) that are monomers (rE$^M$) or stable dimers (rE$^D$) under physiological conditions. We generated stable ZIKV rE$^D$ proteins as previously described for DENV E protein by introducing a disulfide bridge (A264C) in the E-domain II interactive region of the homodimer[9,10] (Fig. 1a). The oligomeric state of the purified rE$^M$ and rE$^D$ proteins were confirmed by protein gel electrophoresis (Fig. 1b). The stable rE$^D$, but not the rE$^M$, was efficiently recognized by neutralizing mAbs that have dimer-dependent quaternary footprints, such as ZIKV-specific mAbs A9E, G9E, and ZKA-230 and flavivirus cross-reactive mAbs C8 and C10 (Fig. 1c).

**ZIKV rE$^D$, but not rE$^M$, induces neutralizing antibodies**. After confirming that rE$^D$ was a dimer displaying native quaternary epitopes, we immunized C57BL/6 mice with 5 μg rE$^M$, 5 μg rE$^D$, 5 μg rE$^M$+alum, or 5 μg rE$^D$+alum and analyzed ZIKV-specific IgG and neutralizing antibody titers over time (Fig. 2a). In the absence of alum, rE$^M$ and rE$^D$ induced minimal IgG titers after the initial immunization, but this response increased upon boosting (Fig. 2b, c). In the alum adjuvanted groups, high IgG titers were observed even after the prime alone, with minimal change upon boosting (Fig. 2b). Although rE$^M$ elicited ZIKV-binding antibodies in the absence of alum, these antibodies failed to neutralize ZIKV (Fig. 2d). In contrast, non-adjuvanted rE$^D$ elicited ZIKV-neutralizing antibodies (Fig. 2d). The addition of alum adjuvant elevated neutralizing antibody levels in both groups, but neutralizing titers were significantly higher after rE$^D$ immunization compared with rE$^M$. Our results demonstrate that, while both antigens stimulated ZIKV-specific IgG, the rE$^D$ stimulated antibodies that are more potently neutralizing compared with the rE$^M$ stimulated antibodies.

**ZIKV rE$^D$ stimulates antibodies that target complex epitopes**. We next performed experiments to determine if antibodies stimulated by rE$^M$ and rE$^D$ antigens recognized different epitopes on the ZIKV E protein. Many flavivirus type-specific antibodies recognize relatively simple epitopes on domain III of the E protein (EDIII), whereas other type-specific antibodies recognize more complex quaternary structure epitopes displayed on E protein homodimers or higher order structures that cover the viral surface. Most EDIII-specific neutralizing antibodies have been isolated from mice immunized with inactivated flavivirus antigens[11–13]. In contrast, most quaternary epitope directed neutralizing antibodies have been isolated from people infected with wild-type flaviviruses[7,14–17]. In fact, recent studies demonstrate that neutralizing antibodies in

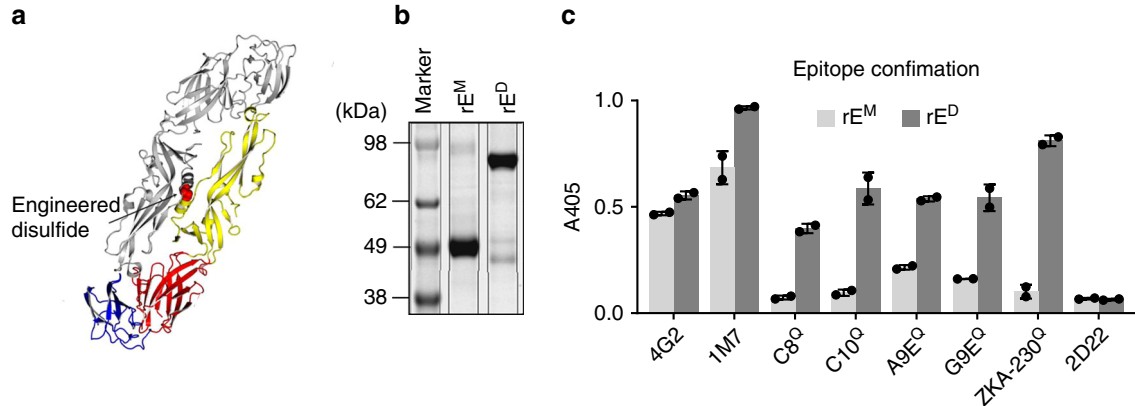

**Fig. 1** Expression pf ZIKV rE monomers and rE homodimers. **a** The soluble ZIKV rE-monomer (rE$^M$, aa1–404) was expressed and purified from mammalian cells. The soluble stable ZIKV rE-homodimer (rE$^D$) was generated by an A264C substitution, resulting in a disulfide bridge in the E-domain II dimer interphase (yellow). **b** Purified rE$^M$ and rE$^D$ were analyzed by SDS-PAGE and exhibited predicted molecular weights of ~49 kDa (monomer) and ~98 kDa (dimer). **c** Binding of ZIKV specific (A9E, G9E, and ZKA-230) and flavivirus cross-reactive (4G2, 1M7, C8, and C10) mAbs to ZIKV rE$^M$ and rE$^D$ was analyzed. Binding of mAbs that have a quaternary footprint (Q) was only observed for rE$^D$. The DENV2 specific mAb 2D22 was used as a negative control. Results from $n = 2$ independent experiments are shown. Error bars represent standard deviation. Source data are provided as a Source Data file

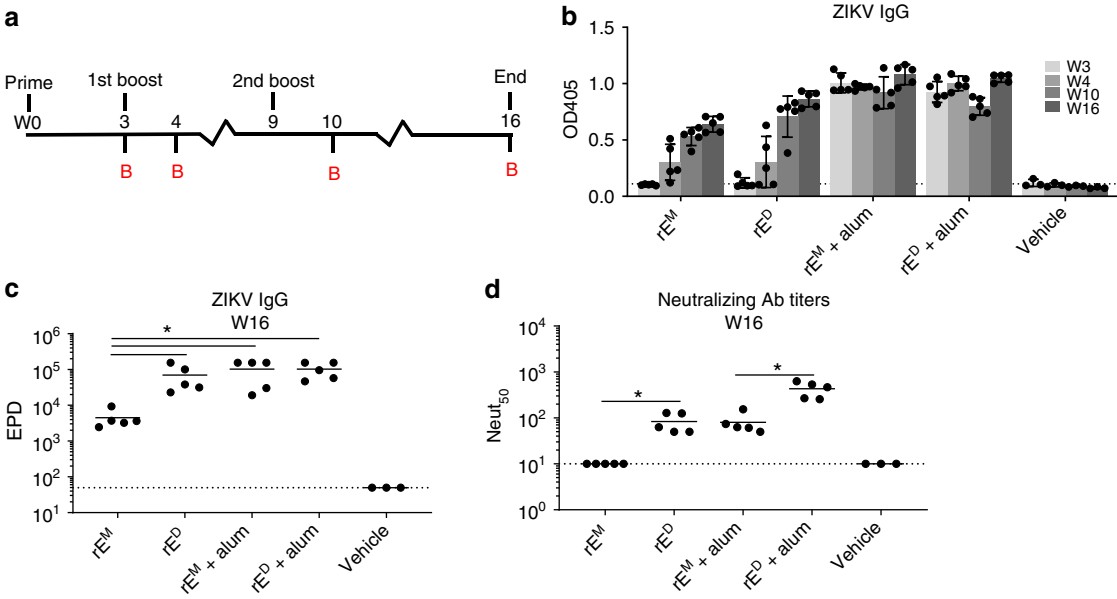

**Fig. 2** ZIKV rE$^D$ induces neutralizing antibody responses. **a** Mice were primed with 5 μg of ZIKV rE$^M$ or rE$^D$ (+/−alum) and boosted with similar doses at weeks 3 and 9. Mice were bled (**b**) at indicated time points (n = 5 for rE$^M$, rE$^D$, rE$^M$+alum, and rE$^D$+alum, and n = 3 for vehicle control). **b** ZIKV-specific single point dilution (1:50) IgG titers were determined by virus capture ELISA. **c** ZIKV-specific endpoint dilution (EPD) titers were determined at week 16 by virus capture ELISA; EPD is the serum dilution at which detection signal equals background levels. **d** ZIKV-neutralizing activity at week 16 was determined on Vero cells and expressed as the serum dilution at which 50% of the virus was neutralized (Neut$_{50}$). Statistical differences were determined by one-way ANOVA followed by a Tukey's test (*$p < 0.05$). Data points represent individual mice. Error bars represent standard deviation. Source data are provided as a Source Data file

people exposed to flavivirus infections or effective live attenuated vaccines predominantly recognize quaternary epitopes, with simple EDIII epitopes accounting for only a minor fraction of neutralizing antibodies[18–22]. To compare levels of EDIII directed antibodies in animals immunized with rE$^M$ or rE$^D$, we used recombinant ZIKV-EDIII conjugated to magnetic beads to deplete mouse immune sera of EDIII-binding antibodies (Fig. 3a). We tested control-depleted and EDIII antibody depleted immune sera to estimate the levels of binding (Fig. 3b) and neutralizing antibodies (Fig. 3c) in each sample targeting epitopes on EDIII. In mice immunized with rE$^M$, ~70% (rE$^M$ group) and ~60% (rE$^M$+alum group) of total ZIKV-specific IgG recognized epitopes on EDIII (Fig. 3b). On the other hand, significantly less EDIII-specific antibodies were induced after immunization with rE$^D$ (~30% for rE$^D$ and ~20% for rE$^D$+alum). The antibody binding patterns were also reflected in the ZIKV neutralization assays. The majority of neutralizing antibodies in the rE$^M$+alum group were lost following depletion with EDIII antigen (Fig. 3c). In contrast, no loss of neutralization was observed in the animals immunized with rE$^D$ or rE$^D$+alum after the removal of EDIII-binding antibodies (Fig. 3c). We conclude that the rE$^M$ antigen induces an EDIII focused antibody response that is poorly neutralizing, whereas rE$^D$ antigens stimulate robust neutralizing antibody responses that target epitopes distinct from simple EDIII epitopes.

We recently isolated two ZIKV-specific strongly neutralizing human mAbs (A9E and G9E) from a person who was infected with ZIKV in 2016[23]. These antibodies map to complex epitopes on domain I (A9E) and domain II (G9E) of ZIKV E protein[23]. Moreover, antibody blockade assays using these mAbs and human immune sera indicated that the epitopes targeted by these mAbs also are targeted by antibodies in immune serum from Zika patients but not dengue patients[23]. We used a blockade of binding (BOB) assay to test if mice immunized with rE$^M$ or rE$^D$ developed antibodies that recognize A9E or G9E epitopes on ZIKV. We also tested if rE$^M$ or rE$^D$ antigens stimulated

antibodies that targeted the epitope of human mAb EDE C10[17] which binds to a E dimer-dependent epitope that is conserved between DENV and ZIKV. Sera from mice immunized with rE$^M$ did not block binding of A9E, G9E, and C10 mAbs to ZIKV (Fig. 3d). In contrast, sera from mice immunized with rE$^D$ antigen efficiently blocked binding of A9E and G9E (~80%) and partially blocked binding of EDE C10 (~35%) to ZIKV. These results demonstrate that the rE$^D$ antigen and natural ZIKV infections stimulate antibodies that target similar complex epitopes on the virion. The rE$^M$ fails to induce similar antibodies and directs the antibody response to simple epitopes on ZIKV EDIII.

**ZIKV rE$^D$ induces protective antibodies in mice.** Next, we investigated if the immune sera of mice vaccinated with rE$^M$+alum and rE$^D$+alum could protect mice from lethal ZIKV challenge. We immunized wild-type mice as previously described (Fig. 2), and transferred heat-inactivated immune sera to Ifnar1$^{-/-}$ mice which are highly susceptible to ZIKV. Ifnar1$^{-/-}$ mice were challenged with 1000 FFU of ZIKV H/PF/2013, 1 day post serum transfer. Mice were monitored for weight loss and ZIKV induced disease, and mice were bled at days 0 (before virus challenge), 4, and 14 (Fig. 4a). The potently neutralizing mAb G9E was used as a positive control for protection against ZIKV infection[23]. Mice that received G9E or serum from rE$^D$+alum immunization were completely protected against weight loss and disease signs. In contrast, mice receiving rE$^M$+alum serum exhibited significant weight loss and hindlimb paralysis (Fig. 4b). At 4 days post challenge, viremia levels in the rE$^D$+alum group were significantly reduced compared with the rE$^M$+alum and vehicle groups (Fig. 4c), and all groups cleared viremia at day 14. We also measured ZIKV-specific IgG titers (Fig. 4d) and neutralizing antibody titers (Fig. 4e) at days 0 and 14. Mice receiving rE$^M$+alum and rE$^D$+alum immune sera showed similar ZIKV-specific IgG titers at day 0, which were

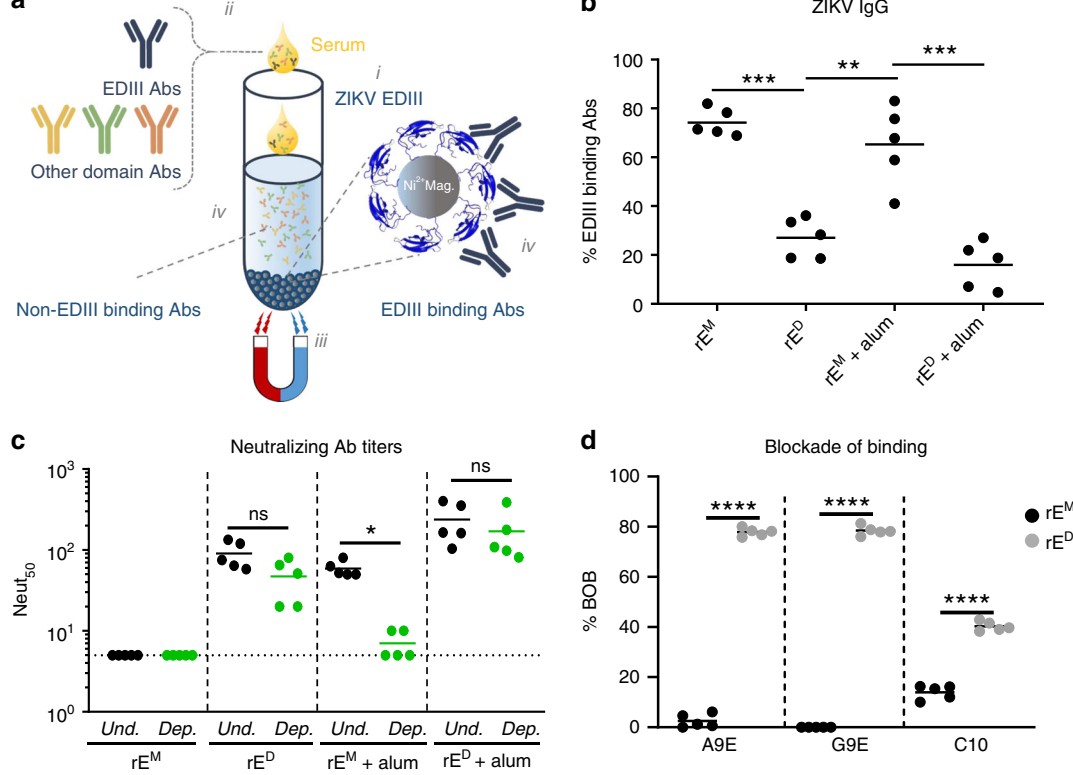

**Fig. 3** ZIKV rE[D] stimulates antibodies that target complex epitopes on the virion. **a** Method for depleting EDIII-binding antibodies from mouse immune sera. Recombinant ZIKV E-domain III (EDIII, His-tagged) was coupled to nickel beads[i] then incubated with immune sera[ii]. Magnetic pull down removes nickel beads and EDIII-binding antibodies[iii], leaving behind sera depleted from EDIII-binding antibodies[iv]. **b** The level of EDIII-binding antibodies in the serum of mice immunized with the indicated antigens is shown as a percentage of the total level of ZIKV-specific IgG, as measured by ELISA of EDIII-depleted and control-depleted serum. **c** The neutralizing activity of EDIII-depleted (green) and undepleted sera (black) was determined and expressed as the dilution at which 50% of the virus was neutralized ($Neut_{50}$). **d** A blockade of binding (BOB) assay was used to evaluate if mice immunized with rE[M] (black) or rE[D] (gray) developed antibodies that blocked the binding of A9E, G9E, and EDE C10 human mAbs. Data points represent individual mice. Statistical differences were determined by one-way ANOVA followed by a Tukey's test ($^*p < 0.05$). Source data are provided as a Source Data file

marginally higher for both groups at day 14. On the other hand, neutralizing antibody titers were higher at day 14 in the rE[M] +alum group, likely caused by an endogenous immune response elicited against replicating virus in these animals (Fig. 4c).

## Discussion
The magnitude, sudden onset and severe clinical manifestations of ZIKV epidemics highlight the need for safe and effective vaccines for key patient populations including pregnant women. While effective live attenuated vaccines have been developed for several flaviviruses, they are contraindicated in pregnant women. Live vaccines typically are ineffective during the first year of life because maternal antibodies interfere with vaccine replication. It has been difficult to formulate tetravalent live attenuated DENV vaccines that stimulate balanced and effective responses to each serotype because of differential replication of vaccine virus components. Subunit vaccines may address some of these challenges, but progress has been slow and subunit vaccines have not performed well in non-human primate models[24]. Our studies offer a potential explanation for the poor performance of subunit vaccines and a strategy for developing more effective flavivirus vaccines. Our results demonstrate that a critical aspect of rE antigen design is to develop E proteins that display complex quaternary epitopes that recapitulate antigen presentation on the surface of infectious virus particles. Here we demonstrate that ZIKV rE[M] and rE[D] subunit antigens induce antibody responses

that differ in functionality and domain focus. ZIKV rE monomers stimulated an EDIII biased, poorly neutralizing IgG response. In contrast, ZIKV rE dimers induced strongly neutralizing and protective antibodies that also blocked the binding of strongly neutralizing human mAbs to quaternary epitopes on ZIKV. While finer mapping studies are required to comprehensively map the specificity of rE[D] stimulated antibodies, our studies suggest that by designing the E protein to be a stable homodimer, it is possible to induce strongly neutralizing and protective antibodies that mimic properties of protective antibodies induced by natural infection. It is possible that inactivated whole virus or virus-like particle (VLP) vaccines will also stimulate antibodies to quaternary structure epitopes. An advantage to using rE[D]-protein subunits over inactivated virion or VLP vaccines is the absence or prM protein in subunit vaccine but not virus or VLP vaccines. PrM antibodies are non-neutralizing and some studies suggest that enhance the replication of flaviviruses[25]. Additional studies are needed to compare the immunogenicity, efficacy and safety of ZIKV rE-dimer, inactivated virus and VLP vaccines. The strategy we describe here for ZIKV E protein may also be applicable for developing subunit vaccines against other medically significant flaviviruses.

## Methods
**Cells and viruses**. Vero-81 cells (ATCC CCL-81) were maintained in DMEM medium (Gibco) + 1% non-essential amino acids, 5% fetal bovine serum, streptomycin (100 μg per ml), and penicillin (100 U per ml) at 37 °C with 5% $CO_2$.

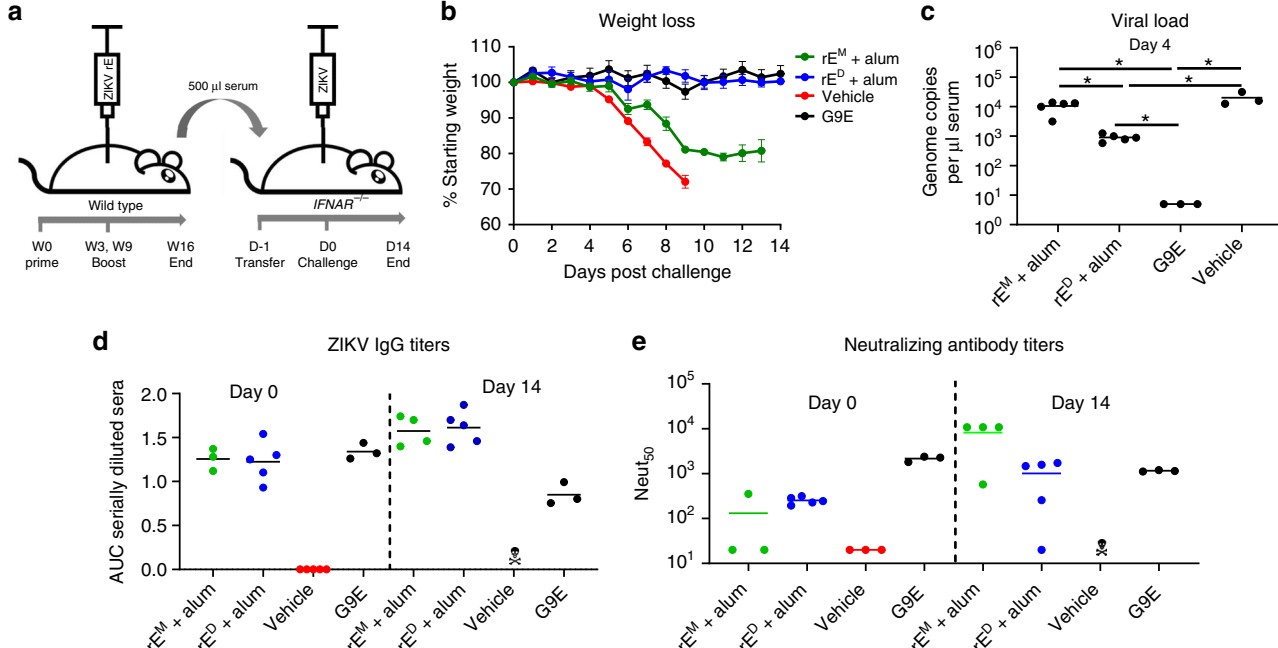

**Fig. 4** ZIKV rE$^D$ induce protective antibodies in mice. **a** Wild-type mice were immunized with rE$^M$+alum or rE$^D$+alum. Five hundred microliters of heat-inactivated immune sera (16 weeks post immunization) (or 200 μg of neutralizing mAb G9E) was transferred by intraperitoneal injection to Ifnar1$^{-/-}$ mice 24 h prior to challenge with 1000 FFU of ZIKV by subcutaneous footpad inoculation ($n = 5$ for rE$^M$ and rE$^D$ and $n = 3$ for controls). **b** Challenged mice were weighed daily for 14 days; weights are shown percent of starting weight. Data represent the mean +/−SEM of 3–5 mice per group and were censored once one mouse in a group died. **c** Viremia was measured at 4 days post challenge by qRT-PCR. **d–e** ZIKV-specific IgG and neutralizing antibody titers were determined for challenged mice prior to challenge (day 0) and at the end of the experiment (day 14). Statistical differences were determined by one-way ANOVA followed by a Tukey's test (*$p < 0.05$). Source data are provided as a Source Data file

Suspension adapted EXPI293 cells (ThermoFisher) were cultured in EXPI293 expression medium (Life Technologies). ZIKV strain H/PF/2013 was grown in Vero cells and titered by focus-forming assay using flavivirus mAb E60[26,27].

**Production of ZIKV rE antigens.** The recombinant ZIKV E (H/PF/2013; aa1–404) monomers (rE$^M$) and stable E dimers (rE$^D$) were expressed in the EXPI293 transient expression system (ThermoFisher) following supplied protocols. The rE$^D$ was created by an A264C substitution in E-domain II and all proteins were equipped with a C-terminal 6 × His-tag. Recombinant proteins were purified as previously described[28]. In short, cell supernatants were concentrated by tangential filtration and proteins were affinity purified using Ni$^{2+}$-chromatography columns. Pooled elution fractions were subjected to size exclusion chromatography and purified protein samples were flash frozen and stored at −80 °C.

**Production of ZIKV EDIII proteins.** ZIKA virus EDIII (H/PF/2013, aa302–403) was expressed as an N-terminal MBP fusion protein equipped with a C-terminal 6 × His-tag in BL21–E. coli. Cells were transformed with a pET21-MBP/ZIKVEDIII expression plasmid and expanded overnight to 100 ml cultures (LB-growth medium, supplemented with 100 μg per ml ampicillin). Overnight cultures were transferred to 1L LB-growth medium and grown to and OD$_{600}$ = 0.6 at 37 °C, shaking at 225 rpm. ZIKV EDIII expression was stimulated by adding 50 mM IPTG and cells were grown overnight at 16 °C with shaking at 225 rpm. Next, cells were pelleted and lysed by high pressure homogenization. ZIKV EDIII proteins were purified from the lysate by Ni$^{2+}$-affinity chromatography. Eluted fractions (50 mM Tris, 10 mM NaCl, 10% glycerol, 10% sucrose, pH 7.0) were pooled and dialyzed overnight at 4 °C in 20 mM Tris, 50 mM NaCl, pH 7.0 and stored at −80 °C.

**Protein analysis.** The oligomeric state of purified rE$^M$ and rE$^D$ proteins was characterized by ELISA and SDS-PAGE followed by Coomassie Brilliant Blue (CBB) staining. For SDS-PAGE, 1 μg of rE$^M$ and rE$^D$ were incubated in denaturing gel loading buffer for 10 min at 95 °C. After centrifugation, proteins were separated by SDS-PAGE and gels (original gel in Source Data File) were incubated with CBB stain (0.1% coomassie blue, 10% acetic acid, 50% methanol). Protein conformations were analyzed by antigen-capture ELISA. ZIKV rE$^M$ and rE$^D$ were captured on Ni$^{2+}$-coated ELISA plates and subjected to binding of ZIKV-specific 2 ng per μl mAbs (A9E, G9E, and ZKA-230)[23] and flavivirus cross-reactive mAbs (4G2 and 1M7)[16,29].

**Mouse immunizations and challenge.** Female C57BL/6 mice were purchased from Jackson Laboratory and used for immunizations at 6–12 weeks of age. Female A129 (B6.129S2-Ifnar1$^{tm1Agt}$/Mmjax) mice were purchased from MMRRC (Mutant Mouse Resource & Research centers supported by NIH) and used in challenge experiments at 7 weeks of age. All mouse experiments were performed under protocols approved by the University of North Carolina Institutional Animal Care and Use Committee, in compliance with ethical and federal regulations (the Public Health Service Policy on Humane Care and Use of Laboratory Animals, Animal Welfare Act, and the Guide for the Care and Use of Laboratory Animals).

For evaluation of ZIKV rE immunogenicity, mice were immunized subcutaneously in the flank with 5 μg soluble rE$^M$ ($n = 5$), 5 μg rE$^M$ + 500 μg alum (Alhydrogel, Invivogen) ($n = 5$), 5 μg rE$^D$ ($n = 5$), 5 μg rE$^D$ + 500 μg alum ($n = 5$), or PBS ($n = 3$). All groups received three immunizations (day 0, 21, and 63), and serum samples were collected on day 21, 28, 70, 77, 84, 91, 98, 105, and 112 by submandibular bleed.

To evaluate the protective activity of rE-induced antibodies, heat-inactivated immune sera were passively transferred to ZIKV susceptible A129 mice. Serum samples (days 70 through 112) of mice immunized with rE$^M$ + alum and rE$^D$ + alum (days 70–112) were pooled and 500 μl of immune sera (per mouse, $n = 5$), or vehicle control serum ($n = 3$) was injected into A129 mice, via intraperitoneal route 1 day prior to ZIKV challenge. The ZIKV-neutralizing mAb G9E[23] was used as a positive control (200 μg/mouse, $n = 3$). Mice were challenged with 1000 FFU of ZIKV (H/PF/2013) by subcutaneous footpad inoculation. Mice were weighed daily for 14 days and monitored for clinical disease signs. Mice losing 30% of their starting weight were humanely euthanized.

**ZIKV rE-induced antibody evaluation.** ZIKV-specific IgG responses were measured by antigen-capture ELISA. Briefly, ZIKV was captured on 1M7 (2 ng per μl)-coated ELISA plates and incubated with 1:50 or serially diluted mouse sera 3, 4, 10, and 16 weeks post immunization. Next, plates were incubated with alkaline phosphatase conjugated anti-mouse IgG (1:2500; Sigma A9044) diluted anti-mouse IgG for 1 h at 37 °C. Wells were developed using alkaline phosphatase substrate and adsorbance was measured at 405 nm. Endpoint dilution (EPD) titers where mice sera signals reached background (vehicle control) levels was determined using GraphPad Prism software.

**Evaluation of neutralizing antibody responses.** Neutralizing antibody titers were determined using a flow cytometry based assay. Vero-81 cells were seeded (25,000 cells per well) and incubated at 37 °C overnight. Immunized mice sera was serially

diluted in OptiMEM (Gibco) supplemented with 2% FBS and incubated for 45 min at 37 °C with the amount of ZIKV previously determined to infect ~15% of the cells. Cells were subsequently washed with OptiMEM and overlaid with the diluted sera/virus mix for 2 h at 37 °C. Next, the cells were washed with growth medium and incubated overnight in growth medium at 37 °C. The next day, cells were washed with PBS and detached from the plate by trypsin (Gibco). Cells were subsequently fixed in 4% paraformaldehyde and washed in permeabilization buffer and blocked with 1% normal mouse serum in perm buffer for 30 min at room temperature. Following incubation, infected cells were detected by staining with Alexa-488 conjugated 1M7 mAb (5 μg per μl) for 1 h at 37 °C. Next, cells were washed in perm buffer and resuspended in 200 μl FACS buffer. The percentage of infected cells were determined using the Guava Flow Cytometer (EMD Millipore) and the neutralizing capacity was expressed as $neut_{50}$ (the dilution where 50% of the virus is neutralized), calculated by GraphPad Prism software.

**Depletion of EDIII-binding IgG from mouse serum.** Bacterially expressed ZIKV EDIII proteins (C-terminal 6 × His-tag) were conjugated to Ni-NTA Magnetic beads (Thermo Scientific) following manufacturers protocol. In short, 1 mg of Ni-NTA beads were washed in equilibration buffer and incubated on a rotator with 60 μg of ZIKV-EDIII or MBP-His (control depletions) for 1 h at 37 °C. Next, the beads were washed in equilibration buffer and divided in two tubes for two rounds of depletions. Mouse immune serum was diluted 1:20 in PBS and incubated with magnetic beads for 1 h at 37 °C on a rotator. Following incubation, the depleted serum was separated from the beads and stored at 4 °C for subsequent IgG ELISA and neutralization assays. The percentage of ZIKV EDIII-binding antibodies was calculated by dividing the EDIII-depleted IgG titers by the control-depleted IgG titers. The percentage of ZIKV EDIII neutralizing antibodies was calculated by dividing the EDIII-depleted $neut_{50}$ by the control-depleted $neut_{50}$.

**Human mAb blockade of binding (BOB) assay.** High-binding ELISA plates were coated with 2 ng per μl 1M7 human mAb in PBS for 1 h at 37 °C. Next, plates were blocked with 3% skim milk in PBS + 0.05% Tween for 30 min at 37 °C. After blocking, plates were washed in PBS + 0.2% Tween and incubated with ZIKV H/PF/2013 for 1 h at 37 °C. The plates were washed and incubated with serially diluted immunized mouse sera for 1 h at 37 °C. Next, the plates were washed and incubated with alkaline phosphatase (AP)-conjugated A9E, G9E or C10 human mAb at 2 ng per μl for 1 h at 37 °C. Following incubation, the plates were washed and developed with AP-substrate and absorbance was measured at 405 nm.

**ZIKV RNA qRT-PCR analysis.** Viral RNA was isolated from 30 μl of mouse sera using the QIAmp Viral RNA Mini Kit (Qiagen) following manufacturer's protocol. RNA was eluted in RNase free MilliQ water and stored at −80 °C. For qRT-PCR, 2 μl of RNA was added to 18 μl of PCR mastermix (10 μl of iTaq universal probes mix + 0.5 μl of iScript advanced reverse transcriptase + 2 μl of forward primer (10 μM; CATGATACTGCTGATTGC) + 2 μl of reverse primer (10 μM; CCTTCCA-CAAAGTCCCTATTGC) + 0.4 μl of probe (IDT; 10 μM; FAM-CGGCATACA-ZEN-GCATCAGGTGCATAGGAG-IBFQ) + 6.5 μl of nuclease-free $H_2O$). Next, samples were analyzed in a Bio-Rad CFX96 Thermocycler according to the following program: 10 min at 50 °C, 2 min at 95 °C (10 s at 95 °C, 20 s at 60 °C) × 40 cycles. RNA was isolated from a known quantity of ZIKV H/PF/2013 to generate a standard curve.

**Reporting summary.** Further information on research design is available in the Nature Research Reporting Summary linked to this article.

## Data availability

The authors declare that all data supporting the findings of this study are available within the paper. The source data underlying the figures are provided as a Source Data file.

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

## Acknowledgements

This work was supported by grants from the NIH-NIAID grants 1-R01-AI107331-01 (PI A. de Silva), U19 AI109784-01 PI J.Ting), US Department of Defense grant W81-XWH-18-2-0035 (PI A. de Silva) and the European Union Zika Plan Research Consortium.

## Author contributions

S.W.M. and A.T. performed the majority of experiments. A.B., J.F., and M.J.M. expressed and purified ZIKV monomer and dimer proteins. C.A.L., H.M.L., and S.T. performed the

animal experiments, S.W.M., S.T., and A.M.dS. designed the study. S.W.M. wrote the paper and all authors reviewed the paper.

## Competing interests
The authors declare no competing interests.
