## [Peer Review File · Nature Communications]

Reviewers' Comments:

Reviewer #1:

Remarks to the Author:

Making a good vaccine for zika virus has been difficult especially it will be used on pregnant women. This study shows the E protein dimeric form could induce much better response than the monomeric version. This is an important development in the vaccine field and provide an alternative to the design of a safe vaccine.

The current work uses prior information from previous work on dengue virus (Slon et al, 2017 and Rouvinski et al, 2017) to design the zika E protein dimers. Different from the previous work, they evaluate in detail the immune response of mice to these dimers in comparison to the monomers. Very clearly the E protein dimers elicit much better protective response than the monomers due to stimulating predominately antibodies that bind across E protein dimer. The paper is very well written and I think the paper is appropriate for Nature Communications.

Major comments are:

(1) Discuss is E protein dimers is a better vaccine than subviral particle vaccine?

(2)When mice is exposed to a mixture of E protein dimers and monomers (which is often the case with other vaccines - subviral particle, inactivated vaccine), does the immune response gets dampened? Maybe the authors could try vaccinating the mice with 1:1 ratio dimer/monomer to see if the neutralization is reduced. If that is the case, then we will know whether a good vaccine should strictly contain pure E protein dimer, either in a protein form or expressed in sub-viral particle ie., inclusion of some monomers will derail the stimulation of protective response.

(3)In Figure 4e, The IFNAR mice, which were injected with antibodies from the WT mice previously inoculated with monomer E protein + alum, on day 14 contains higher levels of neutralizing antibodies than the mice that received serum from dimeric E protein + alum. Can the authors comment on that? Can they also check the viremic level on day 14?

Reviewer #2:

Remarks to the Author:

This manuscript investigates the possibility that dimeric forms of the flavivirus E protein are better immunogens than E monomers. While straightforward, this is an important line of investigation supported by studies demonstrating the importance of E dimers as the fundamental "antigenic unit" of multiple other flaviviruses. The authors constructed a dimeric form of the ZIKV E protein using knowledge arising from prior studies with the dengue virus. Immunization of mice resulted in an antibody response that was superior to monomeric forms in magnitude, and more dramatically when neutralizing activity was assessed. While soluble E protein elicited substantial titers against DIII, the dimeric protein elicited antibodies that recognized the dimer and competed with previously described antibodies that bound a complex quaternary epitope. Passive transfer of ZIKV-immune sera from WT vaccine mice into an immunodeficient model revealed the dimeric form conferred superior protection.

This manuscript contains the appropriate experiments to support the author's arguments and is technically strong.

Minor points

The authors might address the text concerning the state of the ZIKV vaccine field. Are there soluble monomeric proteins in the clinic? The authors suggest that subunit vaccine progress was slow, but this is not a well-supported statement. The ZPIV vaccine (an inactivated subunit vaccine) advanced rapidly

into multiple phase I clinical trials in 2016 and revealed promising immunogenicity data in animal models and humans.

The error bars and the number of independent experiments represented by Figure 1C and 2B need to be clarified.

Point by Point response to reviewers' comments (reviewer comment is in italics and our response in regular text)

Reviewer1: The reviewer commented that our manuscript was “an important development in the vaccine field and provide an alternative to the design of a safe vaccine”. The reviewer had several questions and comments, which we address next.

(1) Discuss if E protein dimer is a better vaccine than subviral particle vaccine?

We have not directly compared E protein dimers to subviral particles. In theory, subviral particles should also induce antibodies to quaternary epitopes because these particles have E dimers and higher order oligomers. We think there are advantages to using E dimers over subviral particles. For example, subviral particles have both E protein and prM proteins. prM protein is not a target of neutralizing and protective antibodies. Some studies suggest prM antibodies might enhance flavivirus replication and disease enhancement (*Luo et al., 2013; BMC Microbiology*). Nevertheless, we plan to conduct direct comparisons of E dimers, subviral particles and inactivated whole virus vaccines in the near future. In the revised manuscript we address this question raised by the reviewer (Page 10, Lines 202-207).

(2) When mice are exposed to a mixture of E protein dimers and monomers (which is often the case with other vaccines - subviral particle, inactivated vaccine), does the immune response get dampened? Maybe the authors could try vaccinating the mice with 1:1 ratio dimer/monomer to see if the neutralization is reduced. If that is the case, then we will know whether a good vaccine should strictly contain pure E protein dimer, either in a protein form or expressed in sub-viral particle ie., inclusion of some monomers will derail the stimulation of protective response.

The reviewer raises the interesting idea that the presence of monomers might dampen the immune response to dimers. We think this question is outside the scope of the current study. However, as we further evaluate monomers and dimers for immunogenicity under different conditions, we will test this interesting idea.

(3) In Figure 4e, The IFNAR mice, which were injected with antibodies from the WT mice previously inoculated with monomer E protein + alum, on day 14 contains higher levels of neutralizing antibodies than the mice that received serum from dimeric E protein + alum. Can the authors comment on that?

As noted by the reviewer, soon after passive antibody transfer to the IFNAR mice, the animals that received immune sera from the monomer and dimer immunized animals had similar levels of neutralizing antibody. However, 14 days after Zika challenge, animals in the rE monomer group had higher neutralizing antibody levels than animals in the rE dimer group. We think this is because the overall higher level of challenge virus replication in the monomer group compared to the dimer group (See Fig. 4C), leads to higher levels of new antibody production in the monomer group compared to the dimer group. This point is now clarified on Page 8, lines 160-162.

4) Can they also check the viremic level on day 14?

We have checked viremia levels at day 14. None of the animals had any viremia by day 14 post challenge. This information has been added to page 8, line 155.

Reviewer 2: This reviewer commented that “this is an important line of investigation supported by studies demonstrating the importance of E dimers as the fundamental “antigenic unit” of multiple other flaviviruses. This manuscript contains the appropriate experiments to support the author's arguments and is technically strong”. The reviewer had two minor points, which we address below.

1) The authors might address the text concerning the state of the ZIKV vaccine field. Are there soluble monomeric proteins in the clinic? The authors suggest that subunit vaccine progress was slow, but this is not a well-supported statement. The ZPIV vaccine (an inactivated subunit vaccine) advanced rapidly into multiple phase I clinical trials in 2016 and revealed promising immunogenicity data in animal models and humans.

ZPIV vaccine is a formalin inactivated whole virus vaccine and not a protein subunit vaccine. The main Zika vaccines in clinical trials are based on live virus, DNA, mRNA and inactivated whole virus vaccines. In the revised paper, we describe the candidates in clinical trials and the fact that protein subunit vaccines are lagging behind (page 2, lines 37-42 and page 10, L186-188).

2) The error bars and the number of independent experiments represented by Figure 1C and 2B need to be clarified.

We have added text to the figure legends to indicate that the results are based on two independent experiments.